# Membranous Expression of Heart Development Protein with EGF-like Domain 1 Is Associated with a Good Prognosis in Patients with Bladder Cancer

**DOI:** 10.3390/diagnostics13193067

**Published:** 2023-09-27

**Authors:** Kohei Mori, Kazumasa Matsumoto, Masaomi Ikeda, Dai Koguchi, Yuriko Shimizu, Hideyasu Tsumura, Daisuke Ishii, Shoutaro Tsuji, Yuichi Sato, Masatsugu Iwamura

**Affiliations:** 1Department of Urology, Kitasato University School of Medicine, 1-15-1 Kitasato, Minami-ku, Sagamihara 252-0374, Kanagawa, Japan; mori2020@med.kitasato-u.ac.jp (K.M.); ikeda.masaomi@grape.plala.or.jp (M.I.); dai.k@med.kitasato-u.ac.jp (D.K.); yulico@med.kitasato-u.ac.jp (Y.S.); tsumura@med.kitasato-u.ac.jp (H.T.); daisukei@med.kitasato-u.ac.jp (D.I.); sato.yuichi@kobal.co.jp (Y.S.); miwamura@med.kitasato-u.ac.jp (M.I.); 2Department of Medical Technology & Clinical Engineering, Gunma University of Health and Welfare, 191-1 Kawamagari-machi, Maebashi-shi 371-0823, Gunma, Japan; tsuji-s@shoken-gakuen.ac.jp; 3KITASATO-OTSUKA Biomedical Assay Laboratories Co., Ltd., 1-15-1 Kitasato, Minami-ku, Sagamihara 252-0329, Kanagawa, Japan

**Keywords:** HEG1, immunohistochemistry, membranous expression, TCGA analysis, bladder cancer, radical cystectomy

## Abstract

Objective: To investigate the correlation between total protein expression of heart development protein with EGF-like domain 1 (HEG1) and clinicopathological characteristics in patients with bladder cancer (BC) after radical cystectomy (RC). Patients and Methods: We retrospectively analyzed data from 110 patients who underwent RC at Kitasato University Hospital. And we prepared an anti-HEG1 monoclonal antibody W10B9, which can detect total HEG1 protein. HEG1 protein expression in tumor cells was evaluated separately for membrane and cytoplasmic staining using immunohistochemistry. Results: Membranous HEG1 expression was associated with absent lymphovascular invasion (*p* < 0.01) and low pT stage (*p* < 0.01). Kaplan–Meier analysis revealed that the membranous HEG1-positive group had significantly long recurrence-free survival (RFS) (*p* < 0.01) and cancer-specific survival (*p* = 0.01). Expression of membranous HEG1 was identified as an independent prognostic factor for RFS (*p* = 0.04). There were no significant differences between cytoplasmic HEG1 expression and clinicopathologic factors including prognosis. Conclusion: The expression of membranous HEG1 could serve as a favorable prognostic indicator in patients with BC treated with RC.

## 1. Introduction

Bladder cancer (BC) is the tenth most common cancer worldwide, with approximately 570,000 new cases in 2020 [1]. It has been reported that both Japanese men and women have relatively high incidence rates of BC (age-standardized rate = 9.6 and 2.2 per 100,000, respectively) [2]. Over recent years, there have been discoveries of several somatic mutations in pivotal genes like the fibroblast growth factor receptor 3, phosphatidylinositol-4,5-bisphosphate 3-kinase, its catalytic subunit alpha, lysine-specific demethylase 6A, and the tumor protein p53. These mutations are believed to influence the prognosis of BC [3,4,5]. Even with the understanding molecular foundations and the progress in integrated treatment methods in BC, approximately 40% of those who undergo a radical cystectomy experience metastases to different organs or lymph nodes within 5 years after radical cystectomy [6].

Heart development protein with EGF-like domain 1 (HEG1) was discovered during the study of a zebrafish heart [7] and is expressed in different tissues involved in various physiological processes such as angiogenesis [8,9], cell–cell junction [10], embryonic development [11,12], and cell proliferation [13] These processes are associated with tumor development and progression, suggesting that HEG1 may play a crucial role in cancer development. Sialylated HEG1 is prominently expressed in many mesothelioma cases, demonstrating its potential as a valuable marker for the differential diagnosis of mesothelioma due to its high specificity [13,14,15,16,17,18]. Conversely, studies exploring the association between HEG1 protein expression and prognosis remain limited, with the exception of hepatocellular carcinoma (HCC) [19]. Therefore, the prognostic significance of HEG1 expression in various other tumor types remains unexplored. Further investigations are needed to reveal the role of HEG1 and its potential implications in patient outcomes across different malignancies.

We previously prepared an anti-HEG1 monoclonal antibody (clone W10B9) that recognizes a part of the peptide sequences in the Ser/Thr-rich region of HEG1 (residues 30–304) and can detect total HEG1 protein (Appendix A). In this study, we investigated the total protein expression of HEG1 and its association with clinicopathological factors and prognosis in patients with BC treated with RC.

## 2. Materials and Methods

### 2.1. Patients and Data Collection

We retrospectively reviewed 143 consecutive patients with BC who underwent RC with pelvic and iliac lymphadenectomy at Kitasato University Hospital (Kanagawa, Japan) from 1990 to 2015. We excluded 10 patients who had histological variants of BC, including 3 with squamous cell carcinoma, 3 with adenocarcinoma, and 4 with small-cell carcinoma, as well as 13 patients who had been previously treated with neoadjuvant chemotherapy and 10 patients who were lost to follow-up. Finally, 110 patients were included in the present study. As a result, a total of 110 patients were suitable for this study. RC was performed on patients confirmed to have muscle-invasive BC also on those with non-muscle-invasive BC who showed no improvement following intravesical treatment [20].

The patients’ characteristics were obtained from their medical records, including age when they underwent surgery, sex, pathological findings (including pT and pN stage), pathology grading, presence of lymphovascular invasion (LVI), past adjuvant chemotherapy (AC), and history of receiving salvage chemotherapy (SC). Tumor grade was assessed using the 1973 World Health Organization grading criteria and the 2002 TNM Classification for Malignant Tumors. The presence of LVI indicates the presence of cancer cells within the endothelial space. AC was considered to perform for patients with >pT2 or those with positive lymph node status. All the patients who underwent either AC or SC received treatment involving platinum-based chemotherapy. The response to chemotherapy was measured using the Response Evaluation Criteria in Solid Tumors (RECIST) version 1.1. We categorized the patients as responders (complete or partial response) or nonresponders (stable disease or disease progression).

This study was approved by the Ethical Committee of Kitasato University School of Medicine and Hospital (Approval Number: B17-010). All participants were approached based on approved ethical guidelines. Patients were informed of their option to opt out through our website and displayed posters, and could refuse to participate or discontinue their participation in the study at any time.

### 2.2. Antigen Purification and Production of Monoclonal Antibody

Antigen of monoclonal antibody W10B9 was prepared as the C-terminal His-tagged recombinant protein that consisted of the N-terminal region of human HEG1 (residues 1–304) (Exon 1–3His). The domain structure of HEG1 is described in ref. (Matsuura R et al. [14] SciRep 8:14251, 2018). The cDNA gene of Exon 1–3His was inserted in pcDNA3.1 (Thermo Fisher Scientific Inc., Tokyo, Japan), and the expression vector was transfected into RK13 cells (RCB0183) (RIKEN BRC, Tsukuba, Japan). The Exon 1–3His recombinant protein was purified by HisTrap excel (1 mL) (Cytiva, Global Life Sciences Technologies Japan K.K, Tokyo, Japan) from 1 L of culture supernatant of stably Exon 1–3His-expressed RK-13 cells. Purified Exon 1–3His (~100 µg/body) was mixed with the same volume of Sigma Adjuvant Systems (Sigma-Aldrich Japan LLC., Tokyo, Japan) and injected into the peritoneal cavity of Balb/c mice. After weekly immunization (4 times), immunized splenocytes were separated and fused with mouse myeloma cell lines (SP2/0-Ag14) (Japanese Collection of Research Bioresources Cell Bank) by the Electro Cell Fusion Generator (ECFG21) (Nepa Gene Co., Ltd., Chiba, Japan). Hybridomas were selected by enzyme-linked immunosorbent assay screening using Exon 1–3His-coated 96-well plates and Western blotting under reducing conditions using mesothelioma cell lines (ACC-MESO4) (RIKEN BRC). After cloning the best reactive cell clones (W10B9 and X9B4), monoclonal antibodies W10B9 and X9B4 were purified with HiTrap protein-G HP (1 mL) (Cytiva) from culture supernatant.

### 2.3. Immunohistochemistry for HEG1

Formalin-fixed and paraffin-embedded tissue blocks representing the most invasive areas of tumor were collected for further investigation. Normal urothelium was harvested from cystectomized specimens. The sections with a thickness of three micrometers were prepared and immunostained using the BOND-MAX automated immunohistochemistry system and the Bond Polymer Refine Detection Kit (DS 9800; Leica Biosystems, Newcastle, UK). The sections were deparaffinized and pretreated using Bond Epitope Retrieval Solution 2 (Leica Biosystems) at 100 °C for 20 min. After washing and peroxidase blocking for 10 min, the sections were rewashed, and immunohistochemistry for HEG1 was performed with 5.26 µg/mL of anti-HEG1 mouse monoclonal antibody (clone W10B9) for 30 min. The preparation method and specificity of W10B9 antibody are shown in the Appendix A. Then, the sections were incubated with BOND Polymer (Leica Biosystems) for 10 min, developed with 3,3′-diaminobenzidine (DAB) chromogen for 10 min, and counterstained with hematoxylin for 5 min. Sections treated with BOND Primary Antibody Diluent (Leica Biosystems) replacing the primary antibody were used for negative controls.

### 2.4. Evaluation of HEG1 Immunostaining

For HEG1, membranous or cytoplasmic staining of tumor cells was considered positive based on previous studies in tumors [19,21]. The staining observed in vascular endothelial cells were used as an internal positive control [11,17]. Although no obvious decrease in HEG1 expression was observed in old archived blocks, the following evaluation method was adapted in consideration of the possibility of reduced antigenicity. The staining intensity of tumor cells was categorized into four groups: 0 = negative; 1 = weaker than vascular endothelial cells; 2 = the same as vascular endothelial cells; and 3 = stronger than vascular endothelial cells. Tumor cells displaying a staining score of either 2 or 3 were considered as positive cells. A tissue consisting of >5% positive cells with the same level or stronger staining intensity than that in vascular epithelial cells was also considered positive. Every immunostained section was meticulously examined by two independent investigators (KM and YS), who were blind to the associated clinical data. Any discrepancies in interpretations were collaboratively reviewed and resolved to reach a mutual agreement.

### 2.5. Kaplan–Meier Plotter Analysis

The relationship between HEG1 mRNA expression and survival across various cancers was examined using the Kaplan–Meier plotter (https://www.kmplot/analysis/ (accessed on 10 May 2023)). We investigated the association between HEG1 mRNA expression levels and overall survival (OS) as well as recurrence-free survival (RFS) in BC. Both hazard ratios (HRs) and log-rank *p*-values were calculated. Patients were divided into two groups according to the best cut-off value of HEG1 mRNA expression.

### 2.6. Statistical Analyses

Age (<65 vs. ≥65 years), tumor stage (≤pT2 vs. ≥pT3), pathological grade (grade 1 and 2 vs. grade 3), and nodal status (N0 vs. N1 and N2) were evaluated as dichotomized variables. The relationship between membranous or cytoplasmic HEG1 expression and clinical-pathological findings was examined using Fisher’s exact test. The Kaplan–Meier method was used to estimate both cancer-specific survival (CSS) and RFS, and their significance was evaluated with the log-rank test. The association between membranous or cytoplasmic HEG1 expression and factors such as pathological stage and grade, presence of LVI, and nodal status was assessed using both univariate and multivariate analyses through the Cox proportional hazards regression model. All statistical calculations were conducted using JMP^®^17 for Windows (SAS Institute Inc., Cary, NC, USA). *p* < 0.05 was considered statistically significant.

## 3. Results

### 3.1. Specificity of Anti-HEG1 Monoclonal Antibodies

The specificity of purified W10B9 and X9B4 for HEG1 was evaluated by Western blotting (Figure 1), and further confirmed by immunostaining using formalin-fixed paraffin-embedded thin sections of a cell block of ACC-MESO4 or HEG1-transfected cells (Figure 2).

### 3.2. Immunohistochemical Staining of HEG1

In non-neoplastic urothelial tissues, HEG1 staining was not observed in the urothelial epithelium (Figure 3A). Vascular endothelial cells were positive even when the tumor cells were negative and used as an internal positive control (Figure 3B). HEG1 staining was observed in the membrane and cytoplasm of the tumor cells (Figure 3C,D).

### 3.3. Association of Clinicopathological Characteristics with HEG1 Protein Expression

Table 1 summarizes the clinicopathological characteristics of all patients and their association with membranous and cytoplasmic HEG1 expression. Positive expression of membranous HEG1 was associated with low pT stage and absence of LVI (*p* < 0.01 and *p* < 0.01, respectively). On the other hand, no significant differences between cytoplasmic HEG1 expression and clinicopathological characteristics were observed. There were no significant differences in chemotherapy responses, irrespective of the localization of HEG1 expression.

### 3.4. Association of Survival Outcomes with HEG1 Protein Expression

Kaplan–Meier analysis revealed that membranous HEG1 expression was associated with a significantly decreased risk of RFS and CSS (*p* = 0.004 and *p* = 0.015, respectively; Figure 4). Cytoplasmic HEG1 expression did not have a significant correlation with RFS and CSS (*p* = 0.249 and *p* = 0.201, respectively; Appendix A). Multivariate Cox regression analysis showed that membranous HEG1 expression, nodal status and adjuvant chemotherapy were independent prognostic factors for RFS (HR = 3.476, 0.392 and 2.501, *p* = 0.04, < 0.01 and 0.01, respectively; Table 2) and that nodal status and adjuvant chemotherapy were independent prognostic factors for CSS (HR = 2.882, 2.712 *p* < 0.01, <0.01; Table 2).

### 3.5. HEG1 mRNA Expression Levels and Survival Outcomes

To compare the present immunohistochemical data, we analyzed HEG1 mRNA expression levels in patients with BC using The Cancer Genome Atlas (TCGA) database (Figure 5). TCGA datasets revealed that the group with high HEG1 mRNA expression had significantly shorter OS, but no significant difference in RFS was observed (*p* < 0.01 and *p* = 0.23, respectively).

## 4. Discussion

This is the first study to investigate the relationship between the expression of HEG1 protein and the clinicopathological background in BC. The results showed that membranous HEG1 expression in tumor cells was associated with long CSS and RFS in patients with BC. In addition, multivariate Cox regression analysis showed that membranous HEG1 expression was an independent predictor for RFS.

HEG1 is a protein encoded by the heart of glass gene that was first discovered in zebrafish [7]. HEG1 protein directly connects with and recruits Krev interaction trapped protein 1 (KRIT1) to endothelial cell junctions to regulate and maintain the organization of junctional molecules, which are critical for vertebrate cardiovascular development [22,23,24]. Other functions of the HEG1 protein such as angiogenesis [8,9], cell–cell junction [10], embryonic development [11,12], and cell proliferation [13] are gradually becoming clear. Kreuk et al. investigated Ras-interacting protein 1 (RASIP1), a membranous protein that is also essential for cardiovascular development. RASIP1 plays a central role in the cell-cell junctions of vascular endothelial cells. They showed that HEG1 binds to the central region of RASIP1, and the deletion of this domain results in an inability to preserve the integrity of cell-cell junctions [10]. These processes are associated with tumor development and progression, so HEG1 may play a crucial role in cancer cell growth.

The relationship of prognosis between HEG1 expression and malignant diseases has been reported in HCC and lung adenocarcinoma [19,21]. Zhao et al. [19] immunohistochemically showed that patients with HCC with high HEG1 expression had aggressive clinicopathological features and shorted OS and disease-free survival than those with low expression. Although the authors found HEG1 expression in the tumor cell membrane and/or cytoplasm, their study did not describe the subcellular localization of HEG1. They further proved that HEG1 plays important roles in HCC invasion, metastasis, and the epithelial–mesenchymal transition by activating Wnt signaling via β-catenin and adenomatous polyposis coli in vitro and in vivo experiments. Zou et al. [21] reported that patients with lung adenocarcinoma with lower HEG1 mRNA expression tended to have a worse prognosis, advanced stage, and high malignancy, indicating that HEG1 has prognostic significance at the clinical level. The results of these two reports were different from the findings of the current study. In the report of HCC, the authors used the same antibody against HEG1, which suggests that the differences in results may be due to the type of cancer cells. In the report of lung cancer, because the authors did not provide enough information about the antibody against HEG1, we could not determine the reason for the difference in results compared to the present study.

In our analysis using the TCGA dataset, we found significantly shorter survival outcomes in the group with high HEG1 mRNA expression. This result was in contrast with the relationship between HEG1 protein expression and survival outcomes that we investigated. In addition, we compared the survival outcomes of cytoplasmic HEG1 cases, and found no significant difference. These results suggest that the biological prognostic impact might depend on the localization of the protein rather than the mRNA expression level. In general, it is expected that higher levels of mRNA expression would result in increased protein expression, but in cancer, these relationships might be altered due to the microenvironment of cancer cells such as unregulated cell growth speed or hypoxia of the tissue [25,26,27]. However, since drugs are directed against proteins, we believe that the relationship with the expression of protein is relatively important.

This study had several limitations. First, it was a retrospective study. Second, the smoking status of the patients was not investigated. Third, the sample size was small, with only 22 patients in the membranous HEG1-positive group. Future large-scale cohort studies are needed to confirm these results. Fourth, while we found that membranous HEG1 could be a potential biomarker for BC, the mechanism of its favorable prognosis remains unknown. Additional studies are required to elucidate the biological function of this protein. Despite these limitations, this present study showed that membranous HEG1 expression is inversely associated with prognosis in patients with BC treated with RC. These results could provide physicians with useful information to establish the next generation of novel treatments.

## 5. Conclusions

Membranous HEG1 expression is associated with a favorable prognosis in patients with BC, indicating that HEG1 is a potential biomarker for BC prognosis. Additional studies are required to elucidate the biological function of membranous HEG1 expression in BC.

## Figures and Tables

**Figure 1 diagnostics-13-03067-f001:**
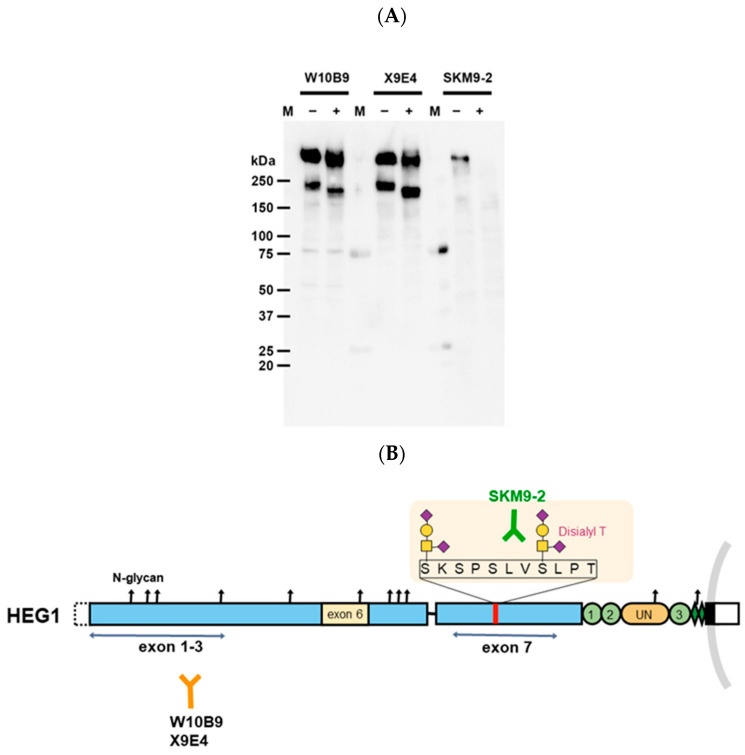
Summary of anti−HEG1 monoclonal antibodies. (**A**) Western blotting using anti−HEG1 monoclonal antibodies. Cell lysates of ACC−MESO4 were treated with the Protein Deglycosylation Mix II (New England Biolabs Japan, Tokyo, Japan), isolated by sodium dodecyl−sulfate poly-acrylamide gel electrophoresis under nonreducing conditions, and then electrotransferred to a PVDF membrane. Then, the membrane was reacted with each monoclonal antibody (clones W10B9, X9E4, or SKM9-2) and analyzed by Western blotting. W10B9 and X9B4, but not SKM9−2, bound to deglycosylated HEG1. M, protein marker (Precision Plus Protein Dual Color Standards (BioRad)); −, nontreated sample; +, glycosidase-treated sample. (**B**) Region of HEG1 molecule recognized by each antibody clone. SKM9-2 antibody recognizes both sialylated O-glycans and peptide sequences in the region shown in (**B**) [14].

**Figure 2 diagnostics-13-03067-f002:**
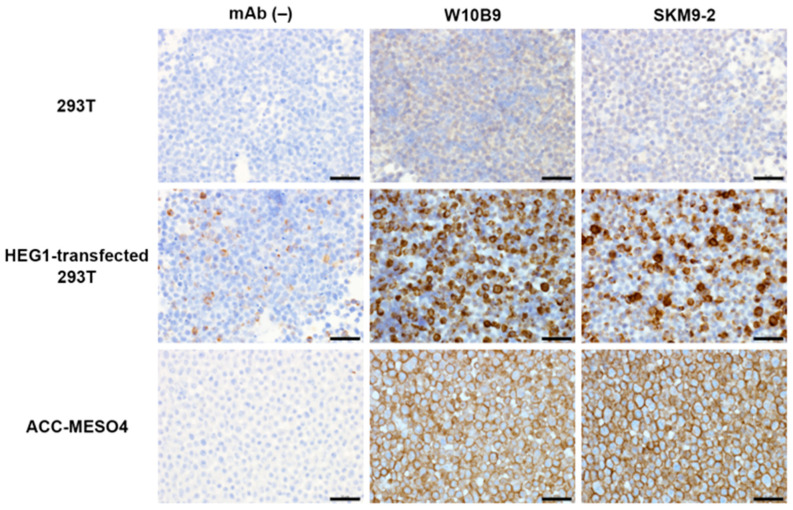
Immunostaining using anti-HEG1 monoclonal antibodies. Cells were collected by centrifugation, fixed in 10% neutral buffered formalin, and embedded in paraffin. Four-micrometer-thick sections of cell blocks were used for immunostaining. HEG1-transfected 293T expressed full-length HEG1. W10B9 and SKM9-2 monoclonal antibodies against HEG1 showed similar stainability. More information about monoclonal antibody SKM9-2 is described in ref. (Tsuji S et al. [13] Sci Rep 7: 45768, 2017). Scale bars, 50 µm.

**Figure 3 diagnostics-13-03067-f003:**
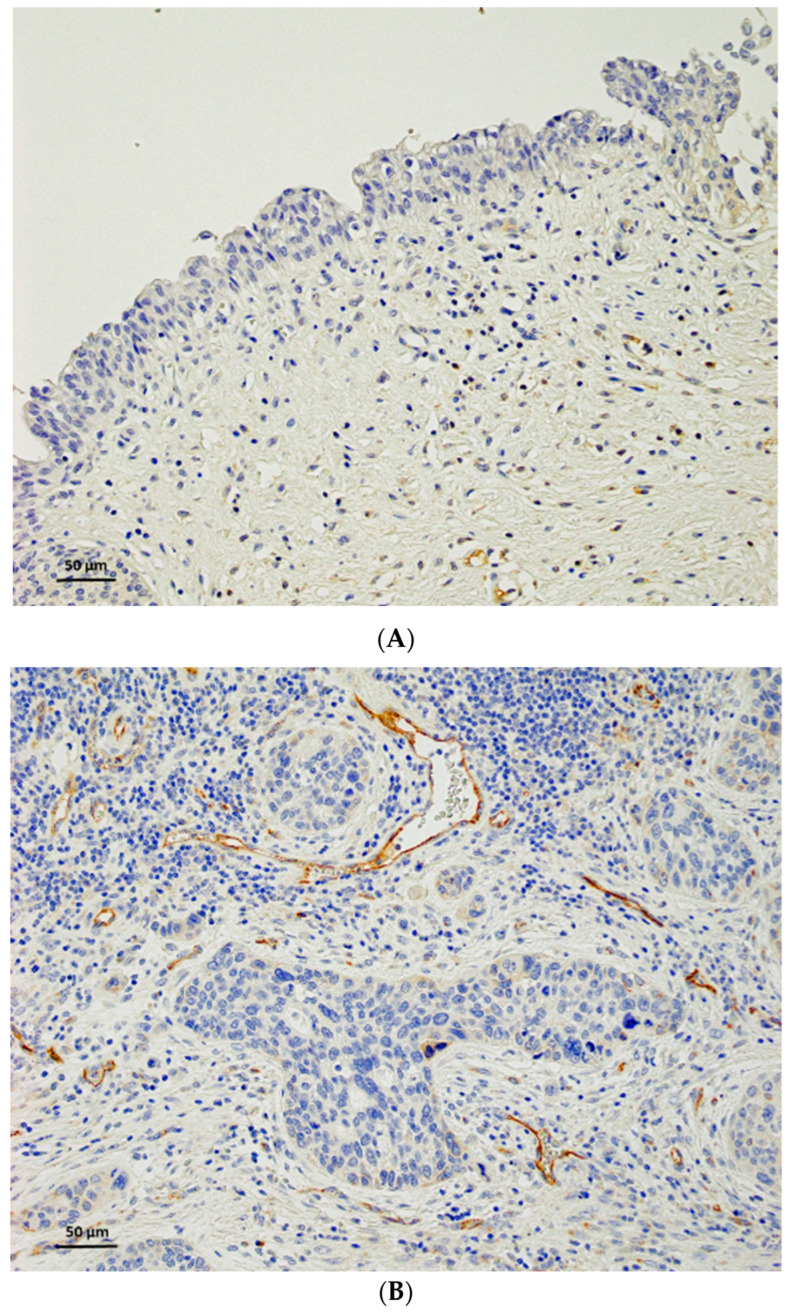
HEG1 expression in non-neoplastic urothelial cells and urothelial carcinoma. (**A**) No or weak staining was observed in non-neoplastic urothelial cells. (**B**) Vascular endothelial cells were positive and tumor cells were negative for HEG1. (**C**) Membranous staining in urothelial carcinoma cells. Scale bar = 50 µm (insert shows high magnification). (**D**) Cytoplasmic staining in urothelial carcinoma cells. Scale bar = 50 µm (insert shows high magnification).

**Figure 4 diagnostics-13-03067-f004:**
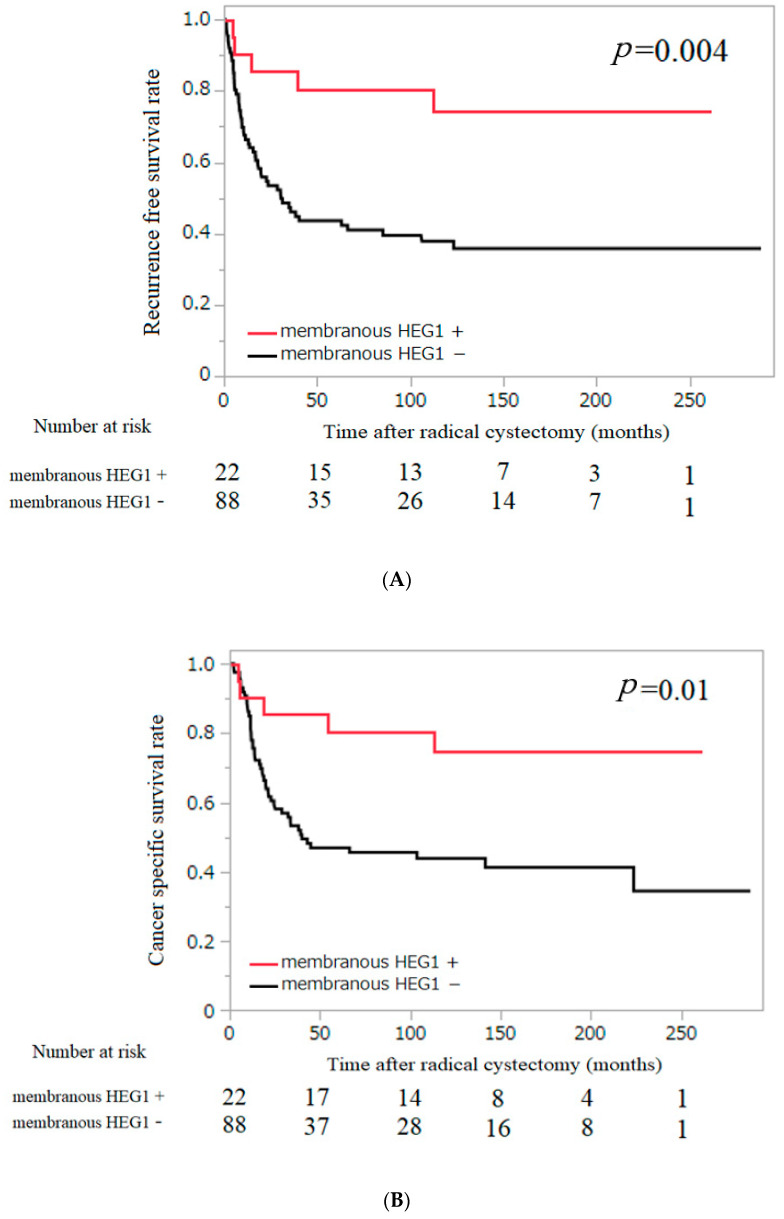
Probability of survival in patients with urothelial carcinoma of the bladder according to membranous HEG1 expression estimated using the Kaplan−Meier method. (**A**) Cancer−specific survival; (**B**) recurrence−free survival.

**Figure 5 diagnostics-13-03067-f005:**
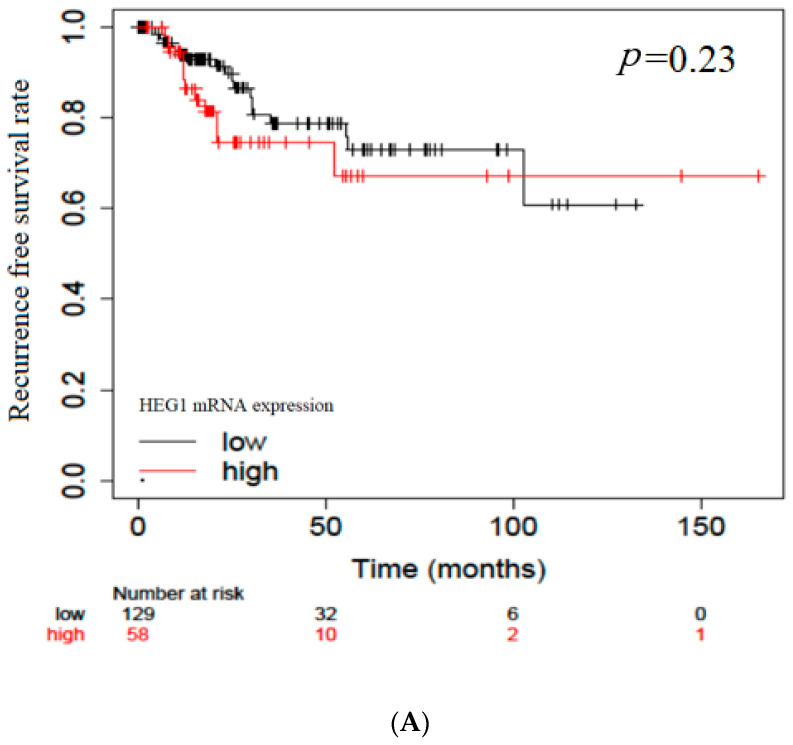
Bioinformatics analysis of survival in patients with urothelial carcinoma of the bladder according to HEG1 mRNA expression estimated using the Kaplan–Meier method (TCGA data). (**A**) Overall survival; (**B**) recurrence-free survival.

**Table 1 diagnostics-13-03067-t001:** Association of membranous and cytoplasmic HEG1 expression with clinical and pathological characteristics.

		Membranous HEG1			Cytoplasmic HEG1		
	No	Positive	Negative	*p*	Positive	Negative	*p*
Total	110	22	88		59	51	
Sex							
Female	22	2	20	0.234	9	13	0.181
Male	88	20	68		50	38	
Age (years)							
≤65	53	11	42	0.849	29	24	0.759
>65	57	11	46		29	27	
pT stage							
≤pT2	47	17	30	<0.001	25	22	0.936
≥pT3	63	5	58		34	29	
Pathological grade							
Grade 1 or 2	33	10	23	0.077	18	15	0.901
Grade 3	77	12	65		41	36	
Lymphvascular invasion							
Absence	39	15	24	<0.001	22	17	0.529
Presence	62	6	56		31	31	
Carcinoma in situ							
Absence	99	21	78	0.691	55	44	0.226
Presence	11	1	10		4	7	
Nodal status							
pN0	80	20	60	0.093	43	37	0.971
pN+	24	2	22		13	11	
Adjuvant chemotherapy							
Yes	13	1	12	0.456	5	8	0.557
No	97	21	76		49	48	
Salvage chemotherapy							
Responder	6	0	6	0.959	2	4	0.303
Nonresponder	21	1	20		3	18	

**Table 2 diagnostics-13-03067-t002:** Multivariate Cox proportional hazards analyses to predict cancer-specific survival and recurrence-free survival in patients with bladder cancer treated with radical cystectomy.

Clinicopathological Variables (No)	Cancer-Specific Survival		Recurrence-Free Survival	
	HR (95% CI)	*p*-Value	HR (95% CI)	*p*-Value
Membranous HEG1 expression Positive (22) vs. negative (88)	0.472 (0.177–1.261)	0.13	0.392 (0.148–1.033)	0.04
Pathological stage ≤pT2 (47) vs. ≥pT3 (63)	1.109 (0.521–2.357)	0.79	1.158 (0.563–2.377)	0.68
Pathological grade Grade 1 or 2 (33) vs. Grade 3 (77)	1.109 (0.539–2.357)	0.84	1.445 (0.742–2.811)	0.43
Lymph node status pN0 (80) vs. pN+ (24)	2.807 (1.399–5.632)	<0.01	3.236 (1.610–6.504)	<0.01
LVI presence Presence (62) vs. absence (39)	1.046 (0.518–2.113)	0.89	0.843 (0.404–1.760)	0.65
Adjuvant chemotherapy Yes (13) vs. No (97)	2.712 (1.325–5.545)	<0.01	2.501 (1.230–5.084)	0.01

CI, confidence interval; HR, hazard ratio; LVI, lymphovascular invasion.

## Data Availability

The datasets used and/or analyzed during the study are available from the corresponding author on reasonable request.

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
