# Peer review of "Membranous Expression of Heart Development Protein with EGF-like Domain 1 Is Associated with a Good Prognosis in Patients with Bladder Cancer"

_diagnostics, 2023, doi:10.3390/diagnostics13193067_

Round 1

Reviewer 1 Report

Comments and Suggestions for Authors

In this article, the authors examined the expression of HEG1 expression in cystectomy specimen.It was shown that a high membranous expression was associated with longer RFS. The concept is novel and interesting, however, there are important points that need to be addressed :

1-Neoadjuvant chemotherapy is the standard of care on MIBC;Why these patients did not receive it?

2- Smoking status might influence the expression of certain epigentics, and affect the survival of these patient(PMID: 37011378). Therefore, Smoking status of these patients should be reported and included in the analysis. 

3-This study included archived tissue for more than 10 years; which might affect the quality of immunohistochemsitry staining of the blocks "Antigen masking" (PMID: 3997553). This might explain weak staining of the old blocks! This need to be addressed in the methodology and the limitation section. 

4- The paper conclude that High expression of membranous HEG has HR of 0.31 of RFS. However, I would encourage the authors to report the effect size to help readers to have a better understanding on the how big is the relationship. 

5- The MVR model didn't account for adjuvant chemo/salvage chemo?  How come AC chemo did not improve RFS?

6-The authors needs to look at the HEG expression between de novo T2 disease and progressive disease from T1. https://doi.org/10.1038/s41598-021-85137-1

Comments on the Quality of English Language

minor syntax errors

Reviewer 2 Report

Comments and Suggestions for Authors

In this work, Mori et al. investigated the association between membranous expression of HEG1 protein and the prognosis in patients with bladder cancer. An anti-HEG1 monoclonal antibody was generated to detect the HEG1 protein specifically. Through quantitative immunohistochemistry analysis, the authors concluded that the expression of membranous HEG1 could be a favorable prognostic indicator in patients with bladder cancer after radical cystectomy. Overall, this is an exciting study. The anti-HEG1 monoclonal antibody developed in this study could be a tool for further investigations on HEG1 in other diseases. However, the authors could consider addressing two minor concerns below to improve this manuscript.

1) A key part of this paper is that immunohistochemistry evaluated HEG1 protein expression in tumor cells separately for membrane and cytoplasmic staining. It would be more convincing if the authors could provide representative images at high magnification to show how the membrane and cytoplasmic staining were identified.

2)  It would be great if the authors could clarify whether the anti-HEG1 monoclonal antibody prepared in this project is a blocking antibody. A functional study on it would be beneficial.

Comments on the Quality of English Language

It is fine.

Round 2

Reviewer 1 Report

Comments and Suggestions for Authors

The authors failed to address my comments, which were not reflected in the updated manuscript.

Comments on the Quality of English Language

NA

Round 3

Reviewer 1 Report

Comments and Suggestions for Authors

All the comments were addressed